# Two-Step Offline Preference-Based Reinforcement Learning with Constrained Actions

## Abstract

Preference-based reinforcement learning (PBRL) in the offline setting has succeeded greatly in industrial applications such as chatbots. A two-step learning framework where one applies a reinforcement learning step after a reward modeling step has been widely adopted for the problem. However, such a method faces challenges from the risk of reward hacking and the complexity of reinforcement learning. To overcome the challenge, our insight is that both challenges come from the state-actions not supported in the dataset. Such state-actions are unreliable and increase the complexity of the reinforcement learning problem at the second step. Based on the insight, we develop a novel two-step learning method called PRC: preference-based reinforcement learning with constrained actions. The high-level idea is to limit the reinforcement learning agent to optimize over a constrained action space that excludes the out-of-distribution state-actions. We empirically verify that our method has high learning efficiency on various datasets in robotic control environments.

## 1 Introduction

Deep Reinforcement Learning (DRL) is a learning paradigm for solving sequential decision-making problems and has rich real-world applications (Sutton & Barto, 2018; Luong et al., 2019; Haydari & Yılmaz, 2020). However, traditional DRL approaches require numerical reward feedback during learning, which in practice can be hard to design or obtain. In contrast, non-numerical preference feedback is more accessible in most cases (Wirth & Fürnkranz, 2013). As a result, preference-based reinforcement learning (PBRL) that only requires preference feedback has become a more realistic learning paradigm (Christiano et al., 2017). Recently, as a special instance of PBRL, reinforcement learning from human feedback (RLHF) that utilizes human preference has drawn much attention and achieved great success in many NLP tasks (Ouyang et al., 2022). Although a majority of current PBRL works focus on the online learning setting (Ibarz et al., 2018; Lee et al., 2021), the PBRL learning in the offline setting is more realistic Rafailov et al. (2023). In the online learning setting, the learning agent continuously interacts with the learning environment to collect new preference data throughout the training process; In the offline learning setting, the agent receives a batch of preference data before training starts. Compared to the online learning setting, the offline learning setting is more accessible as it does not require customized feedback throughout the training process. It is critical to investigate how to make PBRL learning efficient and practical in the offline setting. Particularly, we are interested in the most popular learning framework that is widely applied in various PBRL literature and industrial applications, which we call 'two-step learning.'

**Two-step learning framework:** In a typical two-step learning framework (Ibarz et al., 2018; Christiano et al., 2017), the first learning step is reward modeling, where the learning agent approximates the reward model of the environment that can best explain the observations from the training dataset. The second learning step is a reinforcement learning phase where the learning agent learns a policy based on the reward model acquired in the previous step. There are multiple reasons behind the vast popularity of the two-step learning paradigm: 1. similar to the motivation of inverse reinforcement learning (Arora & Doshi, 2021), the learned environment model provides a succinct and transferable definition of the task; 2. the method is modular in that each phase of learning can be implemented by existing well-developed methods. Despite its popularity, this method faces two main challenges.

- **Reward over-optimization/Pessimistic learning:** Pessimistic learning has always been the key challenge in offline learning settings. Due to the distribution mismatch between the trajectories induced by the policy and that of the dataset, the agent cannot properly evaluate all possible possible policies.

- **Reinforcement learning complexity:** Implementing Reinforcement learning efficiently is known to be challenging Dulac-Arnold et al. (2021). In addition, for preference feedback, the reward model is learned from indirect preference signals rather than actual true rewards. The learned reward model may look very different from the actual reward, making the training step more unstable.

The challenges are empirically verified in Section 5. To solve the first challenge, the most popular approach is to apply a penalty on the evaluated performance of a policy during the reinforcement learning step. The penalty is the KL divergence between a policy and the behavior policy that can best describe the training data distribution. Under the penalty, the agent is encouraged to stay close to the dataset distribution while trying to find a policy that optimizes the learned reward model. However, the reinforcement learning problem in this problem may not be easy to solve due to the second challenge, which is empirically verified in Section 5.

The approaches to solving the second challenge mainly focus on the bandit setting for NLP applications. The bandit setting is a special case of RL that has no state transition. Current methods include direct preference alignment Rafailov et al. (2023) and rejection sampling Liu et al. (2023), both of which cannot be directly applied to the general RL setting. To the best of our knowledge, no existing two-phase learning approach considers tackling both challenges simultaneously.

**Our contributions.** In this work, we propose a novel two-step learning PBRL algorithm PRC that tackles the above two main challenges simultaneously. Our key insight is that both challenges are induced by the same element: the state actions that are out of the dataset distribution. For pessimistic learning, the agent must avoid the policies that are more likely to visit such state-actions. These policies are not well covered by the dataset, and the agent cannot evaluate their performance accurately. Meanwhile, these out-of-distribution state-actions contribute to the complexity of the learning problem. Therefore, our insight is to constrain the action space to exclude all such out-of-distribution state-actions. Based on the insight, the key step in our approach is to constrain the action space to include only the actions with a high probability of being sampled by a behavior clone policy that represents the dataset action distribution. In other words, our method only focuses on the policies that are in the neighborhood of the behavior clone policy. As a result, our method not only inherits the pessimistic idea of behavior regularization but also reduces the complexity of the corresponding reinforcement learning problem. In the experiments, we construct offline PBRL datasets based on standard offline RL benchmark D4RL Fu et al. (2020). The details of dataset construction can be found in Section 5. We empirically verify that on the offline datasets we construct, our approach can always find policies of higher performance than the behavior policies of the dataset, suggesting that the improvement in our PBRL training is reliable.

## 2 RELATED WORK

**Offline Reinforcement learning:** There have been many studies on the standard offline reinforcement learning setting with reward feedback (Levine et al., 2020; Cheng et al., 2022; Yu et al., 2021), in contrast to preference feedback considered in our setting. The key idea behind offline reinforcement learning is pessimism which encourages the learning agent to focus on the policies that are supported by the dataset (Cheng et al., 2022). However, the gap between learning from reward feedback and preference feedback is not clear at present.

**Online PBRL:** Preference-based reinforcement learning is popular in the online learning scenario. Various types of preference models and preference labels have been considered in the tabular case (Fürnkranz et al., 2012; Wirth et al., 2017). Similar to the works that consider the general case, in our work we consider the preference models to be Bradley-Terry models and output binary preference labels (Ibarz et al., 2018; Lee et al., 2021). The two-phase learning approach mentioned in Section 1 is prevalent in the online PBRL setting. Note that directly applying online methods to the offline scenario is also likely to be inefficient even when it is possible (Van Hasselt et al., 2018), as the online learning approaches do not need to be pessimistic.

**Offline PBRL:** Offline preference-based reinforcement learning is a relatively new topic. Zhan et al. (2023) propose an optimization problem following the two-phase learning framework and theoretically prove that the solution to the problem is a near-optimal policy with respect to the training dataset. However, how to solve the optimization problem in practice remains unknown. Concurrently, Zhu et al. (2023) formulates a similar optimization problem under the linear assumption on the environment and proposes a way to solve the problem, but the results cannot be extended to the general case without the linear assumption. Kim et al. (2023) study methods for learning the utility models based on preference provided by humans. They propose to use a transformer-based architecture to predict the utility function behind the preference model. Rafailov et al. (2023) propose a method that directly learns a policy without learning a utility model explicitly. Unlike the setting considered in our work, they focus on the NLP task and require trajectories in each pair to have the same prompt (initial state). Hejna & Sadigh (2023) propose a method that learns a policy and the Q function of the environment instead of the utility function. They also require access to an additional trajectory dataset that only contains trajectories to make their algorithm efficient. Our work focuses on developing a practical and efficient two-phase learning approach for the offline PBRL setting that only has access to a preference dataset with no requirement on the trajectories in the dataset.

**RLHF:** Reinforcement learning from human feedback is a special instance of PBRL that its preference labels are provided by humans. It has been a popular topic recently. Success in various robotic control and NLP taasks have been achieved by fine-tuning pre-trained models through reinforcement learning based on preference feedback from human (Ouyang et al., 2022; Bai et al., 2022). These studies focus on solving specific tasks instead of the general problem considered in our work.

## 3 PRELIMINARY

### 3.1 OFFLINE PREFERENCE REINFORCEMENT LEARNING PROBLEM

We consider an offline preference-based reinforcement learning setting (Zhan et al., 2023). An environment is characterized by an incomplete Markov Decision Process (MDP) without a reward function $\mathcal{M} = (\mathcal{S}, \mathcal{A}, \mathcal{P})$ where $\mathcal{S}$ is the state space, $\mathcal{A}$ is the action space, and $\mathcal{P}$ is the state transition function. A policy $\pi : \mathcal{S} \to \Delta(\mathcal{A})$ is a mapping from a state to a probability distribution over the action space, representing a way to behave in the environment by taking the action at a state. A deterministic policy is a special kind of policy that maps a state to a single action. We use $\Pi_{\mathcal{S},\mathcal{A}}$ to denote the class of all policies, and $\Pi_{\mathcal{S},\mathcal{A}}^D$ to denote the class of deterministic policies. A trajectory of length $t$ is a sequence of states and actions $(s_1, a_1, \ldots, s_t, a_t, s_{t+1})$ where $a_i \in A, s_{i+1} \sim P(\cdot|s_i, a_i) \forall i \in [t]$. A preference model $F$ takes a pair of trajectories $\tau_1, \tau_2$ as input, and outputs a preference $\sigma \in \{\succ, \prec\}$, indicating its preference over the two trajectories, i.e. $\tau_1 \succ \tau_2$ or $\tau_1 \prec \tau_2$. In this work, following the conventions, we consider the preference models that are Bradley-Terry (BT) models (Bradley & Terry, 1952). Specifically, there exists an utility function $u : \mathcal{S} \times \mathcal{A} \to \mathbb{R}$, such that the probability distribution of the output preference of $F$ given $\tau_1, \tau_2$ satisfies:

$$\Pr\{\tau_1 \succ \tau_2\} = \frac{\exp(\sum_{(s,a) \in \tau_1} u(s,a))}{\exp(\sum_{(s,a) \in \tau_1} u(s,a)) + \exp(\sum_{(s,a) \in \tau_2} u(s,a))} \tag{1}$$

We define the performance of a policy $\pi$ on a utility function $u$ as the expected cumulative utility of a trajectory generated by the policy: $\mathbb{E}_{\tau \sim (\pi, \mathcal{P})}[\sum_{s,a \in \tau} u(s,a)]$. An offline dataset $D = \{(\tau_1^1, \tau_2^1, \sigma^1), \ldots, (\tau_1^N, \tau_2^N, \sigma^N)\}$ consists of multiple preference data over different trajectory pairs. We use the term 'behavior policy' to represent how the trajectories are generated in the dataset. A learning agent is given access to the incomplete MDP $\mathcal{M} = (\mathcal{S}, \mathcal{A}, \mathcal{P})$ of the environment and a dataset $D$ whose preferences are generated by a preference model $F$. The learning goal is to learn a policy that has high performance on the utility function $u$ of the preference model $F$, and we say a learning algorithm is efficient if it can learn such a high-performing policy.

## 3.2 TRADITIONAL PPO LEARNING WITH KL-REGULARIZATION

Here, we introduce the prevalent two-step learning framework that has been widely applied for solving PBRL problems Christiano et al. (2017) and is closely related to our algorithm in this work. In general, the algorithm aims at solving the optimization problem below:

$$\arg\max_{\pi \in \Pi_{\mathcal{S},\mathcal{A}}} \mathbb{E}_{\tau \sim (\pi,\mathcal{P})} \sum_{(s,a) \in \tau} [\hat{u}(s,a)] - \alpha \cdot \mathrm{KL}(\pi, \pi_b)$$

$$\text{s.t.} \hat{u} = \arg\min_{u} \mathcal{L}_u(u, D) \tag{2}$$

$$\pi_b = \arg\min_{\pi \in \Pi_b} \mathcal{L}_{\pi}(\pi, D)$$

Here, $\alpha$ is a parameter to control the degree of pessimism. $\hat{u}$ is a learned reward model and $\pi_b$ is a behavior imitation policy. $\mathcal{L}_u(u, D)$ is a loss function to evaluate the quality of the utility model on interpreting the preference labels in the dataset. $\mathcal{L}_{\pi}(u, D)$ is a loss function to evaluate how well the behavior imitation policy reproduces the behavior demonstrated by the dataset.

To solve the optimization problem, the first step is to find the optimal reward model and behavior imitation policy that minimize their corresponding losses. The second step is to use a reinforcement learning algorithm such as PPO to solve for the policy that maximizes the optimization goal. Pessimism is achieved through the KL regularization in the optimization goal, which encourages the policy not to be very different from the behavior imitation policy. Formally, Algorithm 1 below represents a typical way to solve the optimization problem in Eq 2.

---

**Algorithm 1:** Typical two-step training with KL-regularization

**Inputs** : Environment $(\mathcal{S}, \mathcal{A}, \mathcal{P})$, Dataset $D$
1. Train a utility function $\hat{u}$ on $D$ that minimize $\mathcal{L}_u(\cdot, D)$ to approximate the preference model in $D$.
2. Train a policy $\pi_b$ from a class of policy $\Pi_b$ that minimizes $\mathcal{L}_{\pi}(\pi, D)$ to approximate the behavior policy behind $D$.
3. Find the policy $\pi^*$ that optimizes
 $\pi^* = \arg\max_{\pi \in \Pi_{\mathcal{S},\mathcal{A}}} \mathbb{E}_{\tau \sim (\pi,P)}[\sum_{(s,a) \in \tau} \hat{u}(s,a)] - \alpha \cdot \mathrm{KL}(\pi, \pi_b)$
4. Output $\pi^*$

---

# 4 PREFERENCE BASED REINFORCEMENT LEARNING ON A CONSTRAINED ACTION SPACE (PRC)

## 4.1 GENERAL PRC ALGORITHM

First, we introduce our main algorithm PRC. In Alg 2, we formally show the general form of PRC.

---

**Algorithm 2:** Preference Based Reinforcement Learning on a Constrained Action Space

**Inputs** : Environment $(\mathcal{S}, \mathcal{A}, \mathcal{P})$, Dataset $D$
1. Train a utility function $\hat{u}$ on $D$ that minimize $\mathcal{L}_u(\cdot, D)$ to approximate the preference model in $D$.
2. Train a policy $\pi_b$ from a class of policy $\Pi_b$ that minimizes $\mathcal{L}_{\pi}(\pi, D)$ to approximate the behavior policy behind $D$.
3. Construct a clipped action space $\mathcal{A}'$ such that at a state $s$, the clipped action space is
 $\mathcal{A}'|s = \{a : \pi_0(a|s) \geq p\}$.
4. Find the policy $\pi^*$ supported on the clipped action space that optimizes
 $\pi^* = \arg\max_{\pi \in \Pi_{\mathcal{S},\mathcal{A}'}} \mathbb{E}_{\tau \sim (\pi,P)}[\sum_{(s,a) \in \tau} \hat{u}(s,a)]$
4. Output $\pi^*$

---

Here, we abuse the notation of $\pi(a|s)$. If the action space is discrete, then $\pi(a|s)$ represents the probability of the policy to generate the response $a$ at the state $s$. If the action space is continuous,

the $\pi(a|s)$ becomes the probability density. In the first step, the learning agent learns a behavior policy that can best imitate the behavior demonstrated in the dataset and then learns a reward model that best interprets the dataset's preference label. This is the same as the first step in the typical two-step learning framework Alg 1. In the second learning step, the agent learns the policy supported on a clipped action space that maximizes the cumulative return on the learned reward model. At this step, the learning agent only considers the high probability actions according to the behavior policy.

Formally, let the incomplete MDP of the environment be $(\mathcal{S}, \mathcal{A}, \mathcal{P})$. Let the behavior policy and reward model learned in the first step be $\pi_0$ and $\mathcal{R}$ respectively. We define a constrained action space $\mathcal{A}'$ based $\mathcal{A}$ and $\pi_0$. Let $\mathcal{A}'|s$ be the set of actions $\mathcal{A}'$ at a state $s$. The constrained action space $\mathcal{A}'$ consists of all actions that have a probability higher than a threshold to be sampled from the behavior policy at a state $s$: $\mathcal{A}'|s = \{a : \pi_0(a|s) \geq p\}$. Then, at the second learning step, the agent solves an RL problem where the MDP is $(\mathcal{S}, \mathcal{A}', \mathcal{P}, \mathcal{R})$.

## 4.2 ANALYSIS

Here, we show that the reinforcement learning step in PRC is essentially optimizing the reward function under a special behavior regularization constraint. Recap the prevalent PBRL framework in Alg 1. The pessimism in the algorithm is achieved by using KL regularization, which encourages the policy to be not very different from the behavior policy. Here, we consider a different regularization as below.

$$C_p(\pi, \pi_b) = \begin{cases} -\infty & \text{if } \exists (s, a) \in (\mathcal{S}, \mathcal{A}), \pi(a|s) > 0, \pi_b(a|s) < p \\ 0, & \text{otherwise} \end{cases}$$

The optimization problem under such a regularization is $\arg\max_{\pi \in \Pi_{\mathcal{S}, \mathcal{A}}} \mathbb{E}_{\tau \sim (\pi, \mathcal{P})} \sum_{(s,a) \in \tau} [\hat{u}(s, a)] - \alpha \cdot C_p(\pi, \pi_b)$. Under such a regularization, the optimal policy can only be supported in a constrained action space where the probability density of the behavior policy is greater than the threshold $p$, as otherwise, it will suffer from an infinite penalty from the regularization.

Next, we discuss why choosing such a regularization is reasonable. First, unlike the soft constraint, such as KL regularization, our regularization is a hard constraint that forces the policy to stay close to the behavior policy. It is conceptually more conservative in that it explicitly decreases the possibility of reward hacking by outputting some policy that is not close to the behavior policy. Second, it reduces the complexity of finding the optimal policy under the regularization. This can make the reinforcement learning step more accessible and reduce the optimization loss at this step.

## 4.3 PRACTICAL IMPLEMENTATION

Here, we introduce a practical implementation of our PRC algorithm, which is later used in our experiments. First, we can train a deterministic policy $\pi_0$ that has a high probability of reproducing the behavior demonstrated by the dataset. The action given by the deterministic policy can be thought of as the center of the distribution of the actual underlying behavior policy of the dataset. Since we have no prior knowledge about the shape of the distribution of the behavior policy, we may only infer that an action is more likely to be sampled by the behavior policy if it is closer to the center of its distribution. Then, we can approximate the space of $\mathcal{A}'$ as a box whose center is at the deterministic policy. The radius of the action space is a hyper-parameter related to the degree of pessimism.

Formally, Alg 3 is a practical implementation for PRC algorithm. Lines 1-2 learn a utility model and a behavior clone policy, which a standard supervised learning approach can achieve. The loss functions are set in Eq 3 below. In section 5, we empirically verify that training on a constrained action space is an efficient pessimistic learning approach, and the reinforcement learning is also much easier on a constrained action space.

$$\mathcal{L}_u(u, D) = - \sum_{(\tau_1, \tau_2, \sigma) \in D} \log \frac{\exp(\sum_{(s,a) \in \tau^*} u(s,a))}{\exp(\sum_{(s,a) \in \tau_1} u(s,a)) + \exp(\sum_{(s,a) \in \tau_2} u(s,a))}$$

$$\mathcal{L}_\pi(\pi, D) = \sum_{(s,a) \sim D} \|\pi(s) - a\|_2 \tag{3}$$

Lines 3-5 implement an RL environment with a constrained action space. For a policy $\pi_{\text{clip}}$ supported on $\mathcal{S}, \mathcal{A}'$, if it outputs an action $a' \in \mathcal{A}'$, then its corresponding actual policy outputs action $f(s, a')$, and the state transition follows $\mathcal{P}(s, f(s, a')) = \mathcal{P}'(s, a')$, where $\mathcal{P}'(s, a')$ is the state transition in Line 5. Therefore, we can search for the optimal clipped policy $\pi^*$ from $\Pi_{\mathcal{S}, \mathcal{A}'}$ on the MDP with state space $\mathcal{S}$, action space $\mathcal{A}'$, state transition $\mathcal{P}'$, and output the corresponding actual policy of $\pi^*$. In practice, we can solve this by applying the SAC or PPO algorithm on the MDP $\mathcal{M}' = (\mathcal{S}, \mathcal{A}', \mathcal{P}', \hat{u})$.

---

**Algorithm 3:** PRC practical implementation

---

**Inputs** : Environment $(\mathcal{S}, \mathcal{A}, \mathcal{P})$, Dataset $D$
**Parameters:** Positive real number $r$
1. Train a utility function $\hat{u}$ that minimizes $\mathcal{L}_u(\cdot, D)$ to approximate the preference model in $D$.
2. Train a deterministic behavior clone policy $\pi_b^0$ that minimizes $\mathcal{L}_\pi(\cdot, D)$.
3. Construct a constrained action space $\mathcal{A}' = \mathbb{R}^N$ where $N$ is the dimensions of $\mathcal{A}$. Each dimension is constrained on $[-r, r]$.
4. Construct a mapping $f : \mathcal{S} \times \mathcal{A}' \to \mathcal{A}$ as $f(s, a') = \text{Proj}_\mathcal{A}(\pi_b^0(s) + a')$
5. Construct a state transition $\mathcal{P}' : \mathcal{S} \times \mathcal{A}' \to \Delta(\mathcal{S})$ as $\mathcal{P}'(s, a') = \mathcal{P}(s, f(s, a'))$
6. Find the optimal policy $\pi^*$ that optimizes $\arg\max_{\pi \in \Pi_{\mathcal{S}, \mathcal{A}'}} \mathbb{E}_{\tau \sim (\pi, \mathcal{P}')}[\sum_{(s,a) \in \tau} \hat{u}(s, a)]$
7. Output $\pi : \pi(f(s, a)|s) = \pi^*(a|s)$

---

## 5 EXPERIMENTS

### 5.1 SETUP

**Dataset:** Following previous studies Hejna & Sadigh (2023), we construct our offline preference dataset from D4RL benchmark Fu et al. (2020), and generate synthetic preference following the standard techniques in previous PBRL studies (Kim et al., 2023; Christiano et al., 2017). Based on the definition of a standard offline preference-based reinforcement learning setting in Section 3, given a reward-based dataset from D4RL, we construct a preference-based dataset through the following process:

1. Randomly sample pairs of trajectory clips from the D4RL dataset. Following previous studies (Christiano et al., 2017), the length of the clip is set to be 20 steps.

2. For each pair of trajectory clips, compute the probability of a trajectory to be preferred based on the reward signals. To ensure consistency between different datasets, the reward signals are regularized to be bound in $[-1, 1]$.

3. For each pair of trajectory clips, randomly generate a preference label through a Bernoulli trial with the probability computed above.

4. Return the preference dataset consisting of the trajectory clip pairs and the corresponding preference labels.

We consider various datasets from D4RL to represent the general learning scenarios. The robot control environments we choose include 'Hopper', 'HalfCheetah', and 'Walker'. The types of trajectories we choose include 'Medium', 'Medium-Replay', and 'Medium-Expert'. To make the number of state-action in the dataset aligned with that of the D4RL benchmark, the total number of trajectory pairs with a preference label is $1 \times 10^6$.

**Baseline Algorithms:** Here, we consider multiple two-step learning algorithms as baselines and a state-of-the-art attack in the related offline PBRL setting, which are listed below:

1. Two-step learning with KL-regularization: We adopt the traditional two-step learning baseline with KL-regularization as introduced in Section 3. To make it a stronger baseline and the comparison fairer, we allow the algorithm to initialize from the behavior clone policy.

2. Naive two-step learning: To show a naive baseline where pessimistic learning is entirely absent, we adopt a naive two-step learning baseline, which is the same as the traditional two-step learning above except that there is no regularization at all during the reinforcement learning step. Note that we also allow the algorithm to initialize from the behavior clone policy in this baseline.

3. Reward modeling followed by standard offline RL algorithms: Another simple yet efficient two-step learning approach for offline PBRL problems is combining reward modeling with a standard reward-based offline RL algorithm. First, a reward model is learned from the preference dataset. Then, the reward model is used to provide scalar reward labels to the state-action in the dataset. Finally, we apply a standard offline RL algorithm IQL Kostrikov et al. (2021) training on the dataset with scalar reward labels.

4. Oracle: The oracle is trained with true rewards instead of preference labels. Here, we apply IQL training on the base D4RL dataset with true reward signals. Note that the information from the reward signals is strictly more than that from the preference signals in this case. The oracle should be considered as an upper bound on the performance of an offline PBRL algorithm.

5. Inverse preference learning (IPL) Hejna & Sadigh (2023): IPL is the state-of-the-art learning algorithm for a related offline PBRL setting that requires a preference dataset and a behavior dataset. To avoid underestimating the performance of IPL, we apply this algorithm to our PBRL setting and allow it to check the behaviors in the full D4RL dataset.

To ensure a fair comparison, all the methods that require reward modeling share the same learned reward model during training.

**Training setup:** To learn a reward model, we follow a standard supervised learning framework. Specifically, we use a multilayer perceptron (MLP) structure for the neural network to approximate the utility model. The neural network consists of 3 hidden layers, and each has 64 neurons. We use a tanh function as the output activation so that the output is bound between $[-1, 1]$.

To train a deterministic behavior clone policy, the neural network we use to represent the clone policy has the same structure as that for the utility model. To train a stochastic behavior clone policy, the network outputs the mean and standard deviation of a Gaussian distribution separately. The network is an MLP with 3 hidden layers each with 64 neurons. The last layer is split into two for the two outputs. For the mean output, we use a linear function for the last layer. For the standard deviation, we use a linear function and an exp activation for the last layer.

In the reinforcement learning step, we use either the SAC algorithm or the PPO algorithm, depending on the dataset. The neural network for the actor is the same as the network for training a stochastic behavior clone policy. The neural network for the critic is the same as the network for training the reward model.

## 5.2 Learning Efficiency Evaluation

Here, we compare the efficiency of PRC against other baseline algorithms on different datasets. To straightforwardly compare the performance of different learning algorithms, we show the performance of the best policy learned by a method during training.

The scores in Table 1 are the standard D4RL score of the learned policies. The results show that the PRC algorithm has high learning efficiency. It is generally more efficient than other baselines and sometimes even competes with the oracle. Our algorithm performs better than other two-step learning baselines initialized from the behavior clone policy. This observation indicates that starting from the behavior clone policy or its neighborhood is not enough to achieve high learning efficiency, and training on the constrained action space is the key to high learning efficiency for PRC.

Next, we empirically analyze why PRC should be an efficient learning algorithm from the aspect of pessimistic learning and reduced reinforcement learning complexity.

| Dataset | Oracle | PRC | Behavior Clone | Naive two-step | KL two-step | IPL | RM |
|---|---|---|---|---|---|---|---|
| HalfCheetah-Medium | 47.3 ± 0.2 | **47 ± 0.5** | 41.7 ± 1.0 | 40.1 ± 0.5 | 41.9 ± 0.1 | 42.7 ± 0.1 | 43.2 ± 0.1 |
| HalfCheetah-Medium-Replay | 46.1 ± 0.1 | **43.3 ± 0.2** | 32.9 ± 9.2 | 31.9 ± 0.8 | 32.9 ± 0.9 | 34.9 ± 3.1 | 40.1 ± 0.7 |
| HalfCheetah-Medium-Expert | 92.1 ± 0.7 | **78.4 ±2.9** | 44.5 ± 3.6 | 40.9 ± 0.2 | 40.4 ± 0.5 | 41.7 ± 1.0 | 48.2 ± 0.8 |
| Hopper-Medium | 76.1 ± 1.2 | **71 ± 7.5** | 51.4 ± 3.9 | 67.5 ± 2.4 | **73.5 ± 4.0** | **72.4 ± 7.4** | 67.2 ± 0.3 |
| Hopper-Medium-Replay | 76.7 ± 5.3 | 47 ± 19.5 | 40.8 ± 8.5 | 49.4 ± 7.4 | **53.1 ± 2.2** | 39.5 ± 17.5 | 32.2 ± 0.4 |
| Hopper-Medium-Expert | 113.1 ± 0.4 | **100.2 ± 9.5** | 46.7 ± 10.9 | 67.3 ± 7.1 | 71.4 ± 10.9 | 76.9 ± 8.1 | 97.1 ± 5.2 |
| Walker2d-Medium | 84.5 ± 0.3 | **84.4 ± 0.8** | 73.5 ± 1.7 | 81 ± 0.8 | 79.5 ± 2.1 | 80.8 ± 1.6 | 81.9 ± 2.5 |
| Walker2d-Medium-Replay | 83.1 ± 2.3 | **87.9 ± 6.1** | 11.6 ± 0.6 | 49.8 ± 7.3 | 45.4 ± 0.1 | 49.3 ± 3.5 | 71.8 ± 6.4 |
| Walker2d-Medium-Expert | 111.5 ± 0.4 | **110.4 ± 0.6** | 95.9 ± 3.1 | 93.4 ± 4.9 | 92.1 ± 2.3 | **107.5 ± 3.0** | 105.7 ± 6.9 |
| Sum Totals | 730.5 | **669.6** | 439 | 521.3 | 530.2 | 545.7 | 587.4 |

Table 1: Comparison between the performance of different learning methods.

### 5.3 PESSIMISM EFFECTIVENESS

Here, we empirically verify that by training on a constrained action space, the PCA algorithm achieves effective pessimistic learning.

In Figure 1, we show some representative examples. For the PRC algorithm, during training, the performance trend of the learned policies measured on the learned reward model is aligned with that measured on the true reward. This suggests that the pessimistic learning in the PRC algorithm is effective: it mainly considers the policies that are supported by the dataset so that the agent can evaluate their relative performance accurately. In contrast, for naive two-step learning and KL-regularized two-step learning, we find multiple cases where the trends can even be opposite when evaluated on the reward model and on the true reward. These results suggest pessimistic learning is not efficient enough in the baseline methods compared to PRC.

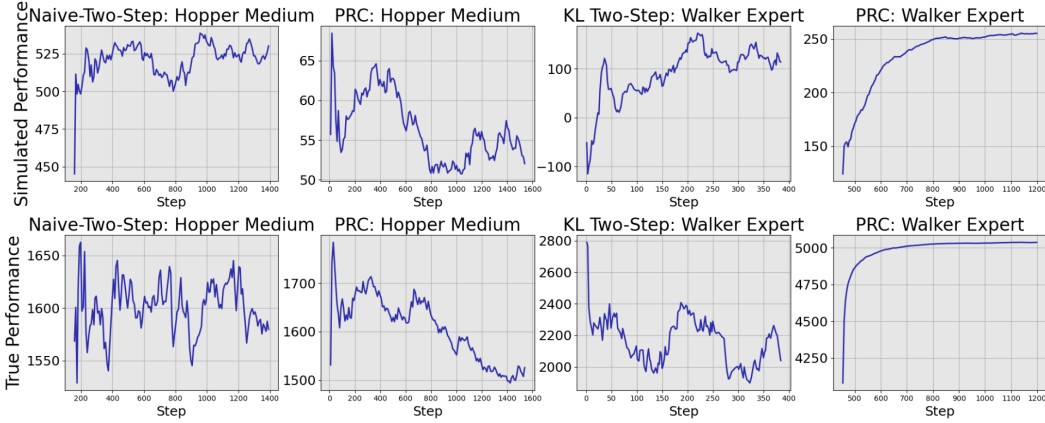

Figure 1: Comparison between the trend of the performance of the learned policies on the learned (simulated performance) and true reward models (true performance) during training. An algorithm is not pessimistic enough if the two trends are not aligned.

### 5.4 REINFORCEMENT LEARNING EFFICIENCY ON CONSTRAINED ACTION SPACE

Here, we empirically verify that reinforcement learning is much easier on the constrained action space. In this case, we focus on the performance of the learned policies evaluated on the learned reward model.

We observe that in most cases, the performance of the learned policies in the PRC method is much higher than in the baseline two-step learning methods. In Figure 2, we show some representative examples. It is clear that while the baseline method struggles with low-performing policies, the PRC method learns policies of much higher performance.

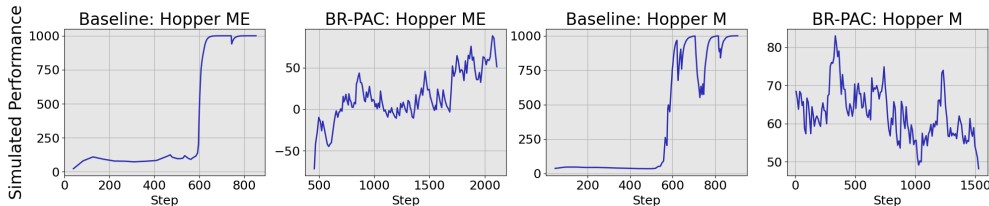

Figure 2: Comparison between the performance of the learned policies on the learned reward models during training. The reinforcement learning complexity is less in a setting if the simulated performance is high.

## 5.5 REINFORCEMENT LEARNING COMPLEXITY IN PBRL

Here, we empirically show that training on a reward model learned from preference signals is harder than that from true reward signals. Given the same D4RL dataset, we train two reward models based on the true reward signals and synthetic preference signals. Then, we train an efficient RL algorithm on the two rewards and compare the learning efficiency in the two cases.

The results in Figure 3 show that when learning on the reward model trained from true rewards, the RL agent can quickly learn some policies that have high performance on the reward model. In comparison, it is hard for the agent to learn well on the reward model from preference signals. This indicates that it is harder to learn from a reward model that is trained on preference signals instead of true reward signals.

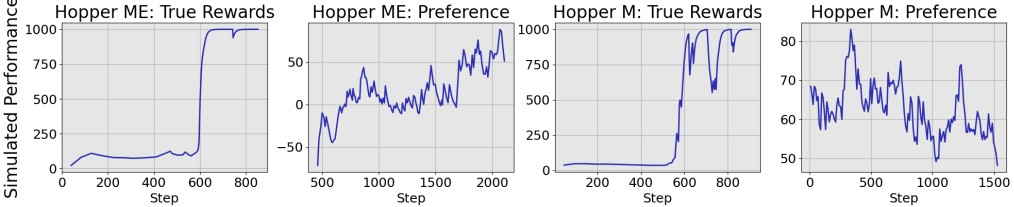

Figure 3: For each dataset, a pair of reward models are trained on the true rewards and preference signals. The same RL algorithm is applied to learn on both reward models. The learning difficulty on a reward model is less if the PPO algorithm can learn better policies according to the reward model.

## 6 CONCLUSION AND LIMITATION

In this work, we propose a novel two-step learning algorithm PRC for the offline PBRL setting. We empirically show that the PRC has high learning efficiency and provide evidence for why it is a more efficient two-step learning algorithm than others. Our framework is limited to the typical offline learning setting and does not answer the question of which trajectories are more worthy of receiving preference labels. Our experimental evaluation is limited to continuous control problems, the standard benchmark in RL studies.

## 7 REPRODUCIBILITY

In the main paper, we explain the setting of the problem we study. The codes we use for the experiments can be found in the supplementary materials.

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
