# OpenReview forum: "Two-Step Offline Preference-Based Reinforcement Learning with Constrained Actions"
_ICLR.cc/2025/Conference — ICLR 2025 Conference Withdrawn Submission_

### Official Review · Reviewer_Ng23 · 2024-10-21

**Soundness:** 2
**Presentation:** 1
**Contribution:** 1
**Rating:** 3
**Confidence:** 4

**Summary:**

The paper proposes PRC (Preference-Based Reinforcement Learning with Constrained Actions) to address challenges in two-step offline preference-based reinforcement learning (PbRL). PRC mitigates reward hacking and instability by constraining the action space to well-represented actions in the dataset. This reduces the complexity of the reinforcement learning phase and improves efficiency. Empirical evaluations on robotic control tasks (D4RL benchmark) demonstrate PRC’s superior performance compared to traditional methods.

**Strengths:**

PRC handles pessimism by constraining actions to those covered in the dataset, improving policy stability.

**Weaknesses:**

1. This paper is indeed incremental, and the contributions (both theoretically and empirically) are not sufficient enough to be presented at this conference. For theory, the authors claim their method can mitigate reward hacking and reduce the complexity of RL, but no quantified analysis is presented. For the experiment, the authors do not offer indicators of the two key contributions.

2. The analysis in Section 4.2 is not enough. More content should be included, such as theoretical analysis of improved efficiency regarding the behavior policy probability density threshold $p$ and the extent to which reward hacking can be mitigated.

3. Further, the experiments need more clarity.

**Questions:**

1. How do the authors estimate the threshold $p$ for the probability density of the behavior policy? Also, as I recognize this threshold as an important parameter, why there are no empirical studies on altering this $p$?
2. I recommend the authors provide explicit expressions (e.g., a math equation) of simulated performance and true performance in the experiment section (Sec 5.1).
3. What is BR-PAC in Figure 2? Why do the authors evaluate the performance of the learned policies on the learned reward models instead of ground-truth reward models? Why does the simulated performance go down in the rightmost subfigure in Figure 2? I do not quite understand the caption, ‘The reinforcement learning complexity is less in a setting if the simulated performance is high.’ I invite the authors to clarify this point.

---

### Official Review · Reviewer_T9c8 · 2024-11-01

**Soundness:** 1
**Presentation:** 2
**Contribution:** 1
**Rating:** 3
**Confidence:** 4

**Summary:**

The paper proposes constraining the action space to make the policy not far from the dataset for offline preference-based reinforcement learning (PbRL). The authors claim that limiting the action space can help reduce reward hacking and RL complexity. Some experiments conducted on D4RL environments show the performance of the proposed method, compared with some simple baseline methods.

**Strengths:**

The paper considers a significant issue in offline PbRL: reward over-optimization, which should be appreciated. To assess the performance of the proposed method, the authors conducted some empirical evaluations on D4RL environments.

**Weaknesses:**

1. The core concept of constraining the action space to mitigate reward hacking and over-optimization is not entirely novel. The modification over conventional offline PbRL is very slignt and lacks considerations. Merely restricting the action space based on the offline dataset could introduce several issues, such as limiting the policy’s ability to explore alternative, potentially optimal actions. This paper does not account for the potential negative consequences of such a simplistic restriction, which may lead to unintended limitations in policy performance and adaptability.
2. The paper lacks a clear structure, with an excessive focus on background information and well-known techniques. The authors should place greater emphasis on detailing their own methods, providing both theoretical insights and empirical analysis to strengthen the contribution of their approach.
3. The experiments are not convincing and comprehensive, as the authors compare their method only to basic baselines in a limited set of simple environments. Additionally, the presentation of the results is unclear. For example, the authors show the performance of the best policy learned by a method during training, which is actually improper. In Table 1, certain values are bolded without explanation. Moreover, the final scores should be reported using normalized rewards rather than raw sums to enable more meaningful comparisons. Furthermore, Figures 1, 2, and 3 lack consistency in their x-axis labeling, making cross-comparisons difficult. Consolidating the results for each task into a single figure and displaying all methods would improve clarity and readability.
4. Overall, this paper does not meet the standards expected for this conference. Significant improvements are needed in both the methodology and experimental evaluation to adequately address the mentioned challenges in offline PbRL.

**Questions:**

Please see the above weakness.

---

### Official Review · Reviewer_Vqgo · 2024-11-04

**Soundness:** 2
**Presentation:** 2
**Contribution:** 2
**Rating:** 3
**Confidence:** 4

**Summary:**

This paper introduces PRC, a two-step learning algorithm for offline PbRL that addresses two major challenges, reward hacking and the complexity of RL with preference feedback. PRC addresses these issues by constraining the agent’s action space to actions with a high probability of being sampled from the dataset’s behavior policy. The authors empirically demonstrate PRC’s effectiveness across various robotic control tasks, showing that it achieves better performance and learning efficiency compared to other two-step learning algorithms.

**Strengths:**

The authors identify and address two key issues in conventional two-step learning algorithms for offline PbRL. The paper is well-motivated and well-organized.

**Weaknesses:**

1. The authors should expand the related work section, as recent key studies on both online and offline PbRL, as well as RLHF for LLMs, are currently missing.

2. Especially, DPPO [1] is a two-step algorithm for offline PbRL that learns a preference predictor followed by direct policy learning through a newly proposed preference score based on policy-segment distance, thereby avoiding reward learning and RL. Additionally, CPL [2] directly learns a policy without both reward modeling and RL. Both approaches address the same challenges as this paper, indicating that this is not the first study to tackle these issues.

3. I am concerned that with unbounded regularization, as mentioned in line 240, optimization might become unstable. Additionally, could excessive conservatism risk resulting in a policy that is unable to achieve high rewards?

4. Further discussion on function $f$ and $\mathcal{P}^\prime$ are necessary to improve clarity and avoid confusion.

5. I recommend including training and evaluation configurations, such as hyperparameters for each method and details on the evaluation process (e.g., number of runs over random seeds), in the appendix for reproducibility.

6. The paper lacks quantitative results for experiments on pessimism effectiveness. Additionally, if these results are based on a single run, I suggest conducting multiple runs and illustrating the findings. This suggestion applies to all the experiments in section 5.3, 5.4, 5.5.

7. Several SOTA baselines in offline PbRL, such as PT [3], DPPO [1], and CPL [2], are missing from the evaluation.

8. Sections 5.4 and 5.5 include observations but provide few discussion of the results.

[1] An, Gaon, et al. "Direct preference-based policy optimization without reward modeling." Advances in Neural Information Processing Systems 36 (2023): 70247-70266.

[2] Hejna, Joey, et al. "Contrastive preference learning: Learning from human feedback without rl." arXiv preprint arXiv:2310.13639 (2023).

[3] Kim, Changyeon, et al. "Preference transformer: Modeling human preferences using transformers for rl." arXiv preprint arXiv:2303.00957 (2023).

**Questions:**

Please see Weaknesses.

---

### Official Review · Reviewer_ANzw · 2024-11-04

**Soundness:** 2
**Presentation:** 2
**Contribution:** 2
**Rating:** 3
**Confidence:** 4

**Summary:**

The author discusses the success of preference-based reinforcement learning (PBRL) in offline settings, particularly in industrial applications like chatbots. A common approach in this area is a two-step learning framework, where reinforcement learning follows a reward modeling step. However, this method faces challenges related to reward hacking and the complexity of reinforcement learning. The author identifies that these challenges stem from state-actions not supported by the dataset, as these state-actions are unreliable and complicate the learning process. To address this issue, the author proposes a novel method called PRC (preference-based reinforcement learning with constrained actions), which limits the reinforcement learning agent to optimizing within a constrained action space that excludes out-of-distribution state-actions. The method is empirically shown to achieve high learning efficiency across various robotic control datasets.

**Strengths:**

1. **Strong Empirical Performance** The empirical results presented in the experiment section demonstrate that PRC significantly outperforms other baseline methods.
2. **Higher Training Efficiency** PRC shows superior learning efficiency compared to other baselines. This efficiency is well-supported by evidence, clearly explaining why PRC is a more effective two-step learning algorithm.
3. **Satisfactory Writing** The overall structure and writing of the paper are satisfactory. It is well-organized, reader-friendly, and generally easy to follow.

**Weaknesses:**

1. **Insufficient Novelty.**
The contributions and novelty of using a two-step learning framework in the PBRL problem are limited. The primary innovation in this work lies in the use of a constrained action space. However, 1) this modification to the original PBRL objective appears to be marginal, and 2) the paper lacks both theoretical and principled justification for the overall performance improvements.
2. **Ambiguous Experimental Results.**
Some of the experimental results and figures are difficult to interpret, leading to confusion. There is a lack of in-depth analysis regarding the model's performance and the underlying reasons for its superior results. A more detailed study is necessary to clarify these points (please refer to the specific question below).
3. **Technical Issues.**
The core of the proposed method is to constrain the action space to include only actions with a high probability of being sampled by a behavior cloning policy. However, it remains unclear how the method guarantees that policies with higher performance than those in the dataset can always be found. This raises concerns about the robustness of the approach.

**Questions:**

1. The PCA algorithm is mentioned in Section 5.3, but it is not explained or referenced earlier in the paper. Could you clarify what the PCA algorithm refers to in this context, and how it relates to the overall methodology?
2. In Section 5.4, it is stated that "the reinforcement learning complexity is reduced when the simulated performance is high." However, I observed that the baseline performance is also quite high. Could you provide further explanation on how we should interpret Figure 2 and the relationship between simulated performance and RL complexity in this context?
3. In the experiments within Section 5.2, your method significantly exceeds the upper bound you set for the Walker2d-Medium-Replay dataset. Given that the action space is constrained, this result seems theoretically unlikely. Could you clarify how your method achieves such results, and whether this outcome is consistent with the constraints imposed by your approach?
4. In Section 4, it appears that Algorithm 3 does not directly address a hard constraint problem during the reinforcement learning process. Instead, it seems to perform a data processing step directly on the dataset. Could you explain how this aligns with or addresses the hard constraints mentioned earlier in the method description?

---

### Note · Authors · 2024-12-04

**Comment:**

We appreciate the insightful comments from the reviewers and will incorporate them in the next version of our work

**Withdrawal Confirmation:**

I have read and agree with the venue's withdrawal policy on behalf of myself and my co-authors.